# Beyond Pharmacology: The Biological Mechanisms of Remote Ischemic Conditioning in Cerebrovascular Disease

**DOI:** 10.3390/biom14111408

**Published:** 2024-11-05

**Authors:** Linhui Qin, Fang Tong, Sijie Li, Changhong Ren

**Affiliations:** Beijing Key Laboratory of Hypoxia Conditioning Translational Medicine, Xuanwu Hospital, Capital Medical University, Beijing 100053, China; qlh@mail.ccmu.edu.cn (L.Q.); tongfang@mail.ccmu.edu.cn (F.T.); lisijie@xwh.ccmu.edu.cn (S.L.)

**Keywords:** cerebrovascular disease, ischemic stroke, chronic cerebral hypoperfusion, remote ischemic conditioning, endogenous protection mechanisms

## Abstract

Cerebrovascular diseases (CVDs), comprising predominantly ischemic stroke and chronic cerebral hypoperfusion (CCH), are a significant threat to global health, often leading to disability and mortality. Remote ischemic conditioning (RIC) has emerged as a promising, non-pharmacological strategy to combat CVDs by leveraging the body’s innate defense mechanisms. This review delves into the neuroprotective mechanisms of RIC, categorizing its effects during the acute and chronic phases of stroke recovery. It also explores the synergistic potential of RIC when combined with other therapeutic strategies, such as pharmacological treatments and physical exercise. Additionally, this review discusses the pathways through which peripheral transmission can confer central neuroprotection. This review concludes by addressing the challenges regarding and future directions for RIC, emphasizing the need for standardized protocols, biomarker identification, and expanded clinical trials to fully realize its therapeutic potential.

## 1. Introduction

Cerebrovascular diseases (CVDs), including ischemic stroke and chronic cerebral hypoperfusion (CCH), are major global health threats and leading causes of disability and mortality, especially in aging populations [1,2,3]. CCH results from pathologically reduced cerebral blood flow (CBF), causing chronic brain dysfunction and potential cognitive impairment due to cell death, impaired autophagy, and inflammation [4,5,6,7,8]. Ischemic stroke, characterized by an obstructed blood supply to the brain, poses treatment challenges due to the strict time windows for interventions like thrombolysis and thrombectomy, which benefit only a limited number of patients [9,10,11,12,13,14]. Thus, there is a pressing need for more effective and safer therapeutic alternatives.

Remote ischemic conditioning (RIC) is a promising non-pharmacological approach aimed at preventing and treating CVDs [15]. By inducing endogenous protective mechanisms, RIC enhances the brain’s tolerance to ischemia and hypoxia, reduces neuronal damage and apoptosis, and improves cerebral blood flow, thereby mitigating the effects of cerebral ischemia and lowering CVD risk [16,17,18]. Numerous studies have demonstrated the significant role of RIC in the prevention and treatment of, as well as the improvement of clinical outcomes for patients with, ischemic stroke and ischemic heart-brain comorbidities [15,19,20]. Specifically, RIC involves the application of blood pressure cuffs to the limbs to induce repetitive short episodes of ischemia, with the intention of protecting distant organs such as the brain and heart from subsequent ischemic injury [21,22,23]. In patients with symptomatic atherosclerotic intracranial artery stenosis (IAS), RIC has been shown to significantly reduce the disease’s recurrence rates and shorten recovery times [24]. Furthermore, clinical research indicates that RIC can markedly decrease the occurrence of ischemia/reperfusion injury [25,26,27]. Recently, a clinical trial demonstrated that chronic remote ischemic preconditioning significantly reduced the incidence of ischemic stroke. This intervention also significantly lowered the frequency of secondary composite outcomes of stroke, transient ischemic attacks, and myocardial infarction [23]. Chen et al. reported that, in adult patients with acute moderate ischemic stroke, RIC significantly increased the likelihood of achieving good neurological function at 90 days compared to usual care [28]. Additionally, a preliminary randomized controlled trial indicated that RIC administered for seven consecutive days in ischemic stroke patients receiving intravenous thrombolysis improved patients’ National Institutes of Health Stroke Scale (NIHSS) scores, thereby facilitating stroke recovery [29].

However, the mechanisms of RIC are still being studied in clinical studies. There have been multiple preclinical studies done in animal models, specifically *rats* and *mice*. These studies have collectively shown that RIC performed prior to the onset of cerebral ischemia reduced the size of infarction by an average of 80% at 48 h [30]. The use of RIC in cerebral ischemia, by instituting occlusion of the femoral artery during the time of reperfusion, showed a 67% reduction in infarct volume 48 h after middle cerebral artery occlusion (MCAO) [31]. There are numerous mechanisms that account for the reduction in cerebral infarct size. Thus, the purpose of this review is to discuss these purported biological processes to understand the physiology and pathology of this treatment for its translation to clinical studies (Figure 1).

## 2. Neuroprotective Mechanisms of RIC on CCH

### 2.1. Improvement of Cerebral Blood Flow (CBF)

The regulation of CBF is crucial for maintaining normal brain function [32]. Insufficient CBF can lead to cognitive decline [33]. This condition is prominently observed in CCH, where there is a significant reduction in CBF [34]. In experimental models, such as the bilateral common carotid artery occlusion (BCAO) model, the CBF has been shown to decrease dramatically, reaching levels as low as 33–58% in critical brain regions including the cortex, white matter, hippocampus, and amygdala. This reduction in CBF is a key factor contributing to the cognitive impairments associated with CCH [35,36]. Therefore, increasing the CBF is considered a potential therapeutic strategy to mitigate cognitive decline induced by CCH. Numerous studies have demonstrated that RIC could enhance CBF in the context of CCH [16,17]. In animal models, it has been observed that the significant decrease in CBF following bilateral carotid artery occlusion (also known as two-vessel occlusion, 2VO) surgery can be reversed through the application of RIC [17]. Furthermore, clinical studies have corroborated these findings, showing that RIC can improve cerebral blood flow in humans [24]. Notably, a one-month regiment of RIC has been found sufficient to reduce cognitive impairment and promote beneficial cerebrovascular remodeling in a model of vascular cognitive impairment and dementia (VCID) [37]. Emerging clinical research also suggests that RIC may improve cognitive function in patients with cerebral small-vessel disease.

Overall, these findings indicate that RIC holds promise as a therapeutic intervention for CCH, primarily through its capacity to improve CBF and potentially alleviate associated cognitive deficits.

### 2.2. Reduction of Apoptosis

Apoptosis, a form of programmed cell death, is a critical factor in endothelial and neuronal cell damage [38]. Stanojlovic et al. observed a large number of apoptotic cells in the 2VO model, noting that such changes were accompanied by increases in levels of the Bax protein and decreases in levels of the Bcl-2 protein [39]. The inhibition of apoptosis may be a way to exert neuroprotection in CCH [39]. Many studies have confirmed that RIC reduces the occurrence of cell death [18,40,41]. An animal study suggests that RIC can prevent the cell death induced by bilateral common carotid artery stenosis (BCAS) after 2 weeks of therapy [18]. However, the protective effect of RIC induced by inhibiting neuronal autophagy needs to be further explored in the clinical setting.

### 2.3. Alleviation of Synaptic Dysfunction

Synaptic dysfunction is a significant contributor to the cognitive impairment caused by CCH [42]. Consequently, many researchers have focused on improving cognitive impairment by reducing the number of synaptic plasticity impairments and increasing the number of synapses [42,43,44]. A decisive study found that RIC alleviates synaptic dysfunction induced by CCH, which in turn improves patients’ cognitive functioning [45]. Additionally, Li et al. demonstrated that RIC enhances the synaptic plasticity in a *mouse* model of CCH by downregulating miR-218a-5p, which in turn upregulates SHANK2 expression. This pathway restoration leads to improved long-term potentiation (LTP) and reduced long-term depression (LTD), thus alleviating the cognitive decline and synaptic dysfunction in models of CCH-induced vascular cognitive impairment [45].

### 2.4. Attenuation of White Matter Damage

White matter hyperintensities, the typical neuroimaging features of cerebral small cerebral vessel diseases, are considered the direct manifestation of CCH [46], and a prominent clinical study reported that RIC can prevent the progression of white matter hyperintensities, indicating that it may serve as a promising a approach to assist drug therapy in improving patients’ cognitive impairment [47]. In an animal study, BCAS resulted in the loss of white matter and myelin basic protein, while RIC was shown to ameliorate this impairment, thus protecting the white matter integrity [18]. Further, Li et al. found that RIC reduces the loss of oligodendrocytes and increases myelin staining and myelin basic protein expression in patients with CCH [40].

### 2.5. Increase of Energy Supply

In the nervous system, the adenosine triphosphate (ATP) and the glucose metabolism play key roles in the supplying of energy for the maintenance of function [48,49]. Cerebral ischemia leads to impaired glucose metabolism [7,20]. A study found that the ATP and glucose levels were significantly reduced in the cerebral cortex of 2VO *rats*, whereas RIC could decrease the ratio of ADP/ATP; increase the glucose content; upregulate the expression of pAMPKα, GLUT1, and GLUT3; and increase the number of GLUT1 and GLUT3 transporters in the cerebral cortical neurons [50].

For the operational methods of RIC, one important variable that must be explored is the beginning time and protocol. Table 1 shows that the popular operational methods for RIC are three cycles of 10 min distal organ I/R for one session per day and four cycles of 5 min distal organ I/R for one session per day. Most studies performed RIC three days and one week after surgery.

## 3. Neuroprotective Mechanisms of RIC on Stroke

RIC has emerged as a promising therapeutic strategy for ischemic stroke, offering neuroprotection through a series of mechanisms that come into play during both the acute phase and the chronic recovery phase following a stroke. This section will discuss the role of RIC in mitigating the damaging factors during the acute phase and its contribution to the restorative processes during the chronic recovery phase of stroke.

### 3.1. Acute Phase Neuroprotection

During the acute phase of stroke, the rapid restoration of blood flow through thrombolysis or thrombectomy is critical to salvaging the ischemic penumbra, the area of the brain tissue that is at risk due to reduced blood supply but still viable. However, reperfusion can also lead to additional injury due to the sudden reintroduction of oxygen and metabolic byproducts, a phenomenon known as ischemia-reperfusion injury. It is in this critical period that RIC can provide significant neuroprotection by modulating several pathophysiological mechanisms.

#### 3.1.1. Reduction of Excitotoxicity

One of the immediate consequences of cerebral ischemia is the disruption of ionic homeostasis, leading to an excessive release of excitatory neurotransmitters, particularly glutamate [41]. This excitotoxicity results in calcium influx into the neurons, activating catabolic enzymes and leading to cell death. A study demonstrated that rapid remote ischemic preconditioning (rRIPC) significantly reduces excitotoxicity by facilitating the regulated release of glutamate from the brain tissue into the peripheral blood. This process decreases the glutamate levels in the ischemic core and penumbra, thereby reducing the neuronal exposure to glutamate toxicity [51]. RIC has been shown to reduce glutamate levels and decrease the calcium overload, thus mitigating excitotoxic neuronal injury [52]. Furthermore, researchers have found that RIC reduces cerebral ischemia-mediated oxidative stress and excitotoxicity by inducing the membrane trafficking of excitatory amino acid transports (EAAT)1 and EAAT2 [53].

#### 3.1.2. Reduction of Apoptosis

Apoptosis is the main cause of neuronal death after ischemia, so rescuing neuronal apoptosis is extremely critical for the treatment of ischemic stroke [54,55]. RIC was found to reduce neuronal apoptosis and decrease the levels of cyclin D1 and CDK6 [56]. In addition, Liu et al. reported that RIC reduced apoptosis by inhibiting the expression of CHOP, a key protein involved in endoplasmic reticulum stress [57]. In a separate mechanism, the activation of the p-TOPK/p-AKT pathway by RIC, as described by Zhao et al., can enhance the expression of antioxidant proteins such as Prx-1, thereby inhibiting apoptosis induced by oxidative stress [58].

#### 3.1.3. Increase in Oxygen Supply

The oxygen supply to the ischemic brain tissue during a stroke is critical to neuroprotection [59]. In cases of ischemic stroke, the obstruction of blood flow due to thrombosis significantly reduces the oxygen delivery to the regions that are distal to the thrombus, leading to the death of many neurons [60]. It has been reported that RIC augments the oxygen delivery capacity of red blood cells (RBCs), which in turn was shown to reduce the infarct volume in a *mouse* model of stroke. RIC significantly increased the levels of 2,3-biphosphoglycerate (2,3-BPG) in erythrocytes, resulting in a higher hemoglobin P50 level, a right-shifted oxygen dissociation curve, and reduced levels of oxygenated hemoglobin in venous blood, indicating improved oxygen delivery and release in tissues [61].

#### 3.1.4. Attenuation of Oxidative Stress

Oxidative stress is key pathological process associated with cerebral ischemia/reperfusion injury. During cerebral ischemia, the lack of oxygen impedes the cellular energy metabolism, resulting in lactic acid accumulation and acidosis. This environment promotes the production of reactive oxygen species (ROS), which contribute to cell death [62,63,64,65]. RIC significantly diminishes the levels of oxidative stress molecules, primarily through the Nrf2/HO-1 pathway, in the MCAO model [66]. Notably, Nrf2, a transcription factor that governs the expression of antioxidant proteins, translocates from the cytosol to the nucleus under oxidative stress, binding to DNA promoters to initiate the transcription of antioxidative genes. This process regulates the expression of antioxidant enzymes and proteins [67]. Additionally, the application of RIC one hour before the occurrence of MCAO reduces the amount of oxidative damage, such as oxidative stress, lipid peroxidation, and oxidative DNA damage in the brain [68]. Consequently, RIC plays a significant role in mitigating oxidative stress induced by ischemia-reperfusion.

#### 3.1.5. Anti-Inflammatory Effects

Remote ischemic conditioning (RIC) exhibits anti-inflammatory effects, which are crucial in the context of stroke-induced inflammation. Following a stroke, dead cells release damage-associated molecular patterns that trigger the production of inflammatory factors [69,70]. This initiates a cascade of events, including the accumulation of immune cells and their infiltration into the brain parenchyma. The levels of microglial activation typically peak 1–4 days post-stroke, which is followed by neutrophil infiltration, which can exacerbate oxidative stress and blood–brain barrier damage [71,72,73]. Therefore, mitigating the inflammatory response that follows acute ischemic stroke is an important target in current research. Preclinical studies have demonstrated that RIC significantly reduces the levels of inflammation mediators such as myeloperoxidase (MPO), tumor necrosis factor-a (TNF-a), interleukin-1β (IL-1β), and IL-6 in MCAO *mice* [11,66,68]. Doeppner et al. reported that delayed RIC reverses ischemia-induced immune inflammation and thus improves neurological recovery [74]. Furthermore, a clinical study reports that RIC relieves inflammatory stress in octo- and nona-genarians with intracranial arterial stenosis [75]. These findings suggest that attenuating the inflammatory response is a crucial target for RIC to ameliorate stroke injury.

#### 3.1.6. Preservation of Blood-Brain Barrier (BBB) Integrity

The blood–brain barrier (BBB) is a physical and metabolic interface that segregates the central nervous system (CNS) from the peripheral circulation, and which is maintained by endothelial cells through tight junctions (TJs) [76]. Disruption of the BBB plays an important role in the development of neurological dysfunction in ischemic stroke [77,78,79]. Previous studies have shown that BBB leakage and the degradation of TJ proteins are increased by up to 3–4-fold in the MCAO model, whereas RIC can ameliorate this increase [80,81]. Recent studies have shown that RIC substantially reduced occurrences of BBB injury, intracerebral hemorrhage, cerebral infarction, and neurological deficits after stroke, even when recombinant tissue plasminogen activator (rtPA) is administrated in a delayed therapeutic time window. Mechanistically, RIC can attenuate rtPA-aggravated BBB disruption after ischemic stroke via reducing the PDGF-CC/PDGFRα pathway [82].

### 3.2. Chronic Recovery Phase Neuroprotection

The chronic phase of stroke, typically defined as the period beyond the initial few weeks following the insult, is characterized by a complex interplay of events aimed at repairing the damaged tissue and restoring function. During this phase, the nervous system’s inherent plasticity is harnessed to facilitate recovery. Remote ischemic conditioning (RIC) plays a crucial role in modulating these restorative processes, which are essential to improving the long-term outcomes after stroke.

#### 3.2.1. Stimulation of Angiogenesis and Arteriogenesis

The restoration of blood flow to the ischemic region is important for the prevention of tissue death after arterial occlusion [83]. After arterial occlusion, the blood vessels respond by increasing the amounts of angiogenesis and arteriogenesis [84,85]. The extent of angiogenesis within the penumbra of ischemic stroke correlates with the patient survival time, suggesting that enhancing these processes may improve the neurological function in patients with ischemic stroke [86]. RIC has been shown to mitigate brain injury, potentially through mechanisms involving vascular neogenesis [87]. Notably, a key study found that delayed RIC also stimulates angiogenesis [74]. The researchers found that rPostC increased the co-expression of BrdU cells with both the endothelial marker CD31 and the immature neuronal marker Dcx 3 months after stroke. Another study showed that RIC treatment significantly bolstered arteriogenesis in the ischemic brains of *rats*, as demonstrated by increased cerebral blood flow, enlarged arterial diameters, and elevated vascular smooth muscle cell proliferation, along with a greater number of leptomeningeal anastomoses. This enhancement in arteriogenesis was closely linked to improved functional outcomes post-stroke [88].

#### 3.2.2. Promotion of Neurogenesis

The central problem with ischemic stroke is neurological dysfunction caused by neuronal damage [89]. Therefore, neurogenesis is a vital process in the recovery of neurological function during the subacute and chronic phases of stroke [90]. It has been reported in some studies that RIC can improve neurogenesis and synaptogenesis in an animal cerebral ischemia model [17,91]. The researchers found that RIC can increase the ratio of EdU+ /DCX+ cells compared the MCAO/R group [87]. A recent study reported that RIC promoted the proliferation and migration of neural stem cells after MCAO [92]. Additionally, pre-conditioning (PreC) combined with post-conditioning (PostC) treatment can improve patients’ neurological function by increasing the number of neural stem cells [93]. In summary, RIC is able to improve neurological recovery in stroke by promoting neurogenesis.

#### 3.2.3. Amelioration of White Matter Injury

The loss of white matter integrity leads to impaired nerve impulse conduction as well as failure of the metabolic and trophic support of axons, resulting in axonal degeneration as well as severe cognitive impairment [94,95]. Clinical studies suggest that white matter repair after stroke is strongly associated with neurological recovery [96], and RIC can reduce the white matter damage in patients with cerebral small vessels [97]. In a prospective and randomized study, Wang et al. speculated that RIC improves cognitive function by ameliorating white matter damage [98]. Therefore, protecting white matter injury may be an important mechanism through which RIC can improve neurological function.

For the operational methods of RIC, one important variable that must be explored is the beginning time and protocol. Table 2 shows that the popular operational methods for RIC are three cycles of 10 min distal organ I/R for one session per day and four cycles of 5 min distal organ I/R for one session per day. Most studies performed RIC immediately after surgery.

## 4. Transmission Pathway of RIC on Neuroprotection

RIC exerts its neuroprotective effects through several action pathways that involve both direct and indirect mechanisms, leveraging the concept of “peripheral transmission for central protection”. These pathways are initiated by RIC and culminate in the activation of endogenous protective mechanisms, leading to a reduction in cerebral infarct size and improvement in the functional outcomes following stroke. By intervening peripherally, RIC triggers systemic responses that translate into central nervous system protection. The following sections will provide an overview of the key pathways through which RIC mediates its protective effects, highlighting this concept.

### 4.1. Peripheral Nervous Transmission

Peripheral nervous transmission is responsible for relaying information between the CNS and the rest of the body, including the muscles and sensory organs [102]. A study showed that the neuroprotective effect of RIC was eliminated when sensory neuroleptics were administered to animals, which represents the neuroprotective mechanism of RIC through the transfer of sensory afferent nerves from the preconditioned limb to the brain [99]. Furthermore, Xiao et al. showed that the electrical stimulation of peripheral nerves carried out by RIC produced neuroprotection in a *rat* model of MCAO [103]. However, peripheral neural transmission pathways are involved in the neuroprotection induced by post- but not pre-ischemic RIC [104]. Surgical resection such as vagotomy also abolished the RIC-mediated protection in *rat* model of myocardial ischemia [105]. These studies collectively suggest the involvement of nervous transmission in the induction of tolerance by RIC.

### 4.2. Immune Modulation

Immune modulation is an important pathological link of ischemic brain injury. RIC has been shown to modulate systemic inflammation by altering several inflammatory pathways. For example, Liu et al. have reported that peripheral immune regulation is one of the important mechanisms through which RIC protects against central nervous injury [100]. Their study indicated that RIC can protect the brain by modulating the levels of various immune cells in peripheral circulation, including CD3+/CD8+ T cells, B cells, and CD3+/CD161a+ NKT cells. Additionally, RIC enhances the presence of inflammation-inhibiting immune cells, such as CD43+/CD172a+ monocytes, which contributes to its anti-inflammatory effects. Recent studies have found that RIC promotes stroke recovery by converting circulating monocytes into a subpopulation of CCR2 proinflammatory monocytes. RIC did not affect the proportions of different phenotypes of monocytes in the spleens of stroke *mice*, but it did decrease the level of Ly-6Clow monocytes and increased the ratio of Ly-6Chigh/Ly-6Clow monocytes to other monocytes in the peripheral blood of stroke *mice* [106].

### 4.3. Exosomes

The protective mechanisms of RIC are not only limited to direct cellular effects but also involve paracrine signaling through exosomes, which are extracellular vesicles that facilitate intercellular communication [107]. Exosomes play a crucial role in RIC’s neuroprotection, as they carry and deliver various biomolecules, including proteins and microRNAs (miRNAs), to distant target tissues [108]. RIC-induced exosomes have been shown to be critical in the translocation of hypoxia-inducible factor-1 alpha (HIF-1α) and other angiogenic and anti-inflammatory factors to ischemic brain tissue. This translocation not only reduces the infarct volume but also promotes neurological recovery by modulating the local tissue environment. The infusion of RIC-derived exosomes has been found to decrease the levels of inflammatory leukocyte factors and increase the infiltration of anti-inflammatory cells in the ischemic brain tissue [109]. Moreover, these exosomes have been shown to promote angiogenesis, which is essential for the restoration of blood flow and nutrient supply to the affected areas [109]. The therapeutic potential of RIC-derived exosomes is further highlighted by studies demonstrating that their protective effects are comparable to those achieved by RIC itself [110].

## 5. Synergistic Effect of RIC Combined with Other Therapies

Remote ischemic conditioning (RIC) holds significant potential for neuroprotection following stroke, and its ability to complement other therapeutic strategies enriches the spectrum of stroke treatment and rehabilitation options. Delving into the interactions of RIC with different interventions, we can gain insights into its amplified benefits for recovery and neuroprotection.

### 5.1. RIC Combined with Atorvastatin

Atorvastatin, recognized for its role in managing cholesterol levels, has shown promising neuroprotective effects by influencing antioxidant and anti-inflammatory pathways post-stroke [111]. Research indicates that preconditioning with atorvastatin can reduce the impact of ischemia-reperfusion injury. The synergistic approach that combines RIC with atorvastatin has demonstrated a marked reduction in reactive oxygen species and neuronal apoptosis, surpassing the effects of either therapy used in isolation [112]. Notably, because of its ease of use, tolerability, affordability, and great neuroprotective potential in models of stroke, atorvastatin combined with RIC is an attractive option for translational research. This combination’s practicality, safety, cost-effectiveness, and substantial neuroprotective potential make it a compelling candidate for further investigation in stroke research.

### 5.2. RIC Combined with Exercise

Physical exercise is increasingly recognized as a valuable rehabilitation strategy for stroke patients. The combination of RIC and exercise has been shown to elicit superior outcomes compared to exercise alone or RIC alone. For instance, sequential RIC followed by exercise enhances neuroplasticity and thereby promotes rehabilitation in ischemic *rats* [113]. Animals treated with RICE (RIC + exercise) had significantly improved functional outcomes after stroke compared to exercise only [91]. RICE, particularly RIC initiation at hour 6 post-reperfusion followed by exercise on day 5, enhanced post-stroke rehabilitation in rats. These rehabilitation groups showed significant improvement in their functional outcomes and levels of synaptogenesis and angiogenesis [114]. Notably, in clinical trials, it was found that RIC may have beneficial effects on the recovery of the lower limb, especially in terms of motor function, by enhancing the levels epidermal growth factor (EGF), [115]. These findings indicate that combining RIC with physical exercise may be more effective in facilitating neuroprotection and rehabilitation in stroke models compared to exercise alone [91].

## 6. Challenges and Future Directions

RIC has emerged as a novel and promising therapeutic strategy for cerebrovascular diseases, including CCH and ischemic stroke. RIC has been shown to be well tolerated in patients with acute ischemic stroke, and it may benefit these patients by improving their clinical outcomes [116]. In patients with intracranial atherosclerosis, long-term repeat RIC can be performed safely and benefit patients by reducing recurrent ischemic stroke and transient ischemic attacks and improving cerebral perfusion status [24,75]. The application of RIC represents a significant shift towards non-pharmacological, preventive approaches that can be applied both in the acute phase and the chronic recovery phase of stroke. The simplicity and non-invasiveness of RIC, coupled with its apparent safety and efficacy, make it an attractive option for clinical translation. Moreover, its potential to be combined with other therapeutic strategies, such as pharmacological interventions and physical exercise, offers a promising avenue for optimizing stroke recovery and rehabilitation.

Despite the promising findings, there are several challenges and areas that require further exploration to fully harness the potential of RIC in clinical practice: 1. Optimization of the RIC protocol: while RIC has demonstrated efficacy, there is a need to standardize and optimize the conditioning protocols, including the duration, frequency, and timing of RIC application, to maximize its therapeutic benefits. 2. Pre- and post-conditioning strategies: pre-conditioning and post-conditioning are two strategic applications of RIC that offer distinct opportunities for neuroprotection in the context of cerebrovascular disease. Pre-conditioning involves the application of RIC prior to the onset of ischemia, aiming to prepare the brain for a potential ischemic event by inducing a state of tolerance. This approach has been shown to be effective in experimental models, where the timing of the ischemic event is controlled. Post-conditioning, on the other hand, is applied after the onset of ischemia and takes advantage of the brain’s innate plasticity during the critical period following a stroke. This strategy has the potential to modulate the ischemic response and promote restorative processes, offering a more clinically feasible window for intervention. 3. The identification of biomarkers: the development of biomarkers to monitor the response to RIC and predict treatment outcomes can help in personalizing therapy and enhancing the efficacy of RIC interventions. 4. Clinical trial expansion: Larger-scale, multicenter, randomized controlled trials are necessary to validate the findings from preliminary studies and to evaluate the long-term safety and efficacy of RIC in diverse patient populations. 5. Combination therapy: further research is needed to explore the synergistic effects of RIC when combined with other established and emerging therapies, such as statins, antioxidants, and rehabilitative exercises, to develop comprehensive treatment strategies. 6. Long-term outcomes: studies should focus on the long-term impacts of RIC on cognitive function, quality of life, and the prevention of recurrent strokes, which are critical for assessing the overall benefit of RIC in real-world settings.

By addressing these challenges and expanding our understanding of RIC through comprehensive clinical trials and translational research, we can bridge the gap between experimental findings and clinical practice, ultimately enhancing the therapeutic potential of RIC for cerebrovascular diseases.

## 7. Conclusions 

In conclusion, the present review underscores the significant potential of RIC as a non-pharmacological therapeutic strategy in the management of cerebrovascular diseases, particularly in the context of chronic cerebral hypoperfusion (CCH) and ischemic stroke. The evidence collated herein highlights RIC’s multifaceted neuroprotective mechanisms, which include the improvement of cerebral blood flow, reduction of apoptosis, attenuation of synaptic dysfunction, and mitigation of white matter damage. Furthermore, RIC has demonstrated the capacity to enhance endogenous protective pathways, modulate immune responses, and stimulate angiogenesis and neurogenesis, which are crucial for stroke recovery. The synergistic effects of RIC when combined with other therapeutic modalities, such as pharmacological treatments and physical exercise, suggest a promising avenue for optimizing patient outcomes. Despite these advancements, challenges remain in the optimization of RIC protocols, identification of predictive biomarkers, and the need for larger-scale clinical trials to solidify RIC’s role in clinical practice. Future research should focus on these areas to fully realize the therapeutic potential of RIC and to develop comprehensive treatment strategies that can effectively combat the devastating impact of cerebrovascular diseases on global health.

## Figures and Tables

**Figure 1 biomolecules-14-01408-f001:**
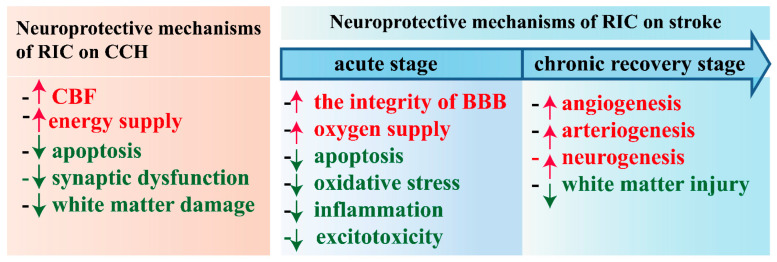
Overview of the biological neuroprotective mechanisms of RIC in CCH and ischemic stroke.

**Table 1 biomolecules-14-01408-t001:** Summarized description of reported studies of RIC on CCH.

Animal	Model	Biological Mechanism	When RIC Was Started	RIC Protocol	References
SD male *rats*	2VO	synaptic plasticity	3 days after 2VO	10 min × 3 cycles, once daily	[45]
SD male *rats*	2VO	CBF, angiogenesis	3 days after 2VO	10 min × 3 cycles, once daily	[17]
C57BL/6J male *mice*	BCAS	CBF, angiogenesis, arteriogenesis, white matter	1 week after BCAS	5 min × 4 cycles, once daily	[37]
C57BL/6 J male *mice*	BCAS	CBF, inflammation, Aβ, cell death, demyelination	1 week after BCAS	10 min × 4 cycles, once daily	[18]
SD male *rats*	2VO	ATP, glucose transport	3 days after 2VO	10 min × 3 cycles, once daily	[50]
SD male *rats*	2VO	demyelination, white matter	3 days after 2VO	10 min × 3 cycles, once daily	[40]

RIC, remote ischemic conditioning; SD: Sprague Dawley; 2VO: 2 vessel occlusion; CBF: cerebral blood flow; BCAS: bilateral common carotid artery stenosis; ATP: Adenosine Tri-phosphate. 10 min × 3 cycles: 10 min of ischemia followed by10 min of reperfusion for three cycles.

**Table 2 biomolecules-14-01408-t002:** Summarized description of reported studies of RIC on stroke.

Animal	Model	Biological Mechanism	When RIC Was Started	RIC Protocol	References
SD male *rats*	MCAO for 2 h	cell apoptosis, endoplasmic reticulum stress	immediately after ischemia	10 min × 3 cycles, once daily	[57]
C57BL/6J male *mice*	MCAO for 70 min	oxygen supply, cell death	immediately after ischemia	10 min × 3 cycles, once daily	[61]
C57BL/6J male *mice*	MCAO for 60 min	brain edema, antioxidant, oxidative stress	immediately after stroke.	5 min × 3 cycles, once daily	[64]
C57BL/6J male *mice*	MCAO for 60 min	oxidative stress, inflammation	immediately after reperfusion	5 min × 4 cycles, once daily	[66]
SD male *rats*	MCAO for 90 min	brain edema, oxidative damage, inflammation, apoptosis.	at 1 h before MCAO	5 min × 4 cycles, once daily	[68]
C57BL/6J male *mice*	MCAO for 60 min	angiogenesis, neurogenesis, immunosuppression,	very delayed remote ischemic post-conditioning, depending on the survival periods of the animals	10 min × 3 cycles, once daily	[74]
SD male *rats*	MCAO for 90 min	brain edema, BBB	immediately after ischemia	10 min × 3 cycles, once daily	[80]
SD male *rats*	MCAO for 90 min	brain edema, BBB	immediately after ischemia	10 min × 3 cycles, once daily	[81]
SD male *rats*	thromboembolic stroke	BBB, cerebral hemorrhage	RIPreC: 7 days before the embolic stroke; RIPostC: immediately after rtPA infusion.	5 min × 4 cycles, twice daily	[82]
C57BL/6J male *mice*	MCAO for 60 min	neurogenesis	at 10 min after MCAO	10 min × 3 cycles, once daily	[92]
SD male *rats*	MCAO for 30 min	brain edema, BBB, inflammation	RIPreC: before stroke	15 min × 3 cycles, once daily	[99]
SD male *rats*	MCAO for 90 min	BBB, inflammation	RIPreC: 1 h before MCAO	5 min × 4 cycles, once daily	[100]
SD male *rats*	MCAO for 90 min	CBF, activates GLP-1R	at 10 min before reperfusion	5 min × 4 cycles, once daily	[101]
SD male *rats*	MCAO for 90 min	arteriogenesis, CBF	immediately after MCAO	10 min × 3 cycles, once daily	[88]
SD male *rats*	MCAO for 2 h	neurogenesis, angiogenesis	3 days after MCAO	10 min × 3 cycles, once daily	[87]

RIC: remote ischemic conditioning; SD: Sprague Dawley; MCAO: middle cerebral artery occlusion; BBB: blood–brain barrier; RIPreC: remote ischemic preconditioning; RIPostC: remote ischemic postconditioning; CBF: cerebral blood flow; GLP-1R: glucagon-like peptide-1 receptor; 10 min × 3 cycles: 10 min of ischemia followed by 10 min of reperfusion for three cycles.

## Data Availability

Not applicable.

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
