# Peer review of "Beyond Pharmacology: The Biological Mechanisms of Remote Ischemic Conditioning in Cerebrovascular Disease"

_biomolecules, 2024, doi:10.3390/biom14111408_

Round 1

Reviewer 1 Report

Comments and Suggestions for Authors

In this article, the authors reviewed the literatures regarding remote ischemic conditioning (RIC) for chronic cerebral hypoperfusion (CCH) and ischemic stroke, summarizing their biological mechanisms. I find this review article to be well-organized and easily comprehensible even for non-specialists in this field. However, I would like to point out a few things to enhance the value of this article.

Major points

1.     [Overall] In this article, the authors described the mechanisms of RIC, the combinatorial effects of RIC and other treatments, and the consequent future directions, referring both pre- and post-conditioning. It may be applicable to discuss the mechanisms by combining insights from pre-conditioning and post-conditioning in chapters 2 and 3, as there are overlapping parts in their mechanisms. However, from a translational perspective, they should be clearly distinguished in chapter 6. For example, applying pre-conditioning to acute ischemic stroke is challenging; however, post-conditioning could be applied with the expectation of therapeutic effects.

2.     [Neuroprotective mechanisms of RIC on stroke] The authors described the reduction of excitotoxicity as a mechanism of RIC for acute phase ischemic stroke. However, the current article does not provide direct evidence suggesting that the reduction of excitotoxicity is the mechanism underlying acute phase neuroprotection in ischemic stroke. Reference #51 used different animal models; therefore, the authors should cite other appropriate references to support this mechanism.

3.     [Neuroprotective mechanisms of RIC on stroke] Regarding chapter 3.2, Chronic Recovery Phase Neuroprotection, some references do not provide mechanistic evidence of RIC for ischemic stroke in the chronic phase. For instance, references #85 and #89 reported results related to angiogenesis or neurogenesis up to 14 days after MCAO and RIC, which do not support neuroprotective mechanisms in the chronic phase. Additionally, since reference #17 utilized animal models for CCH, citing this study to explain the mechanism of RIC for chronic ischemic stroke may be somewhat confusing.

Minor points

4.     [Overall] The abbreviation of ‘2VO’ for bilateral carotid artery occlusion may be a little confusing.

5.     [Challenge and Future Directions] To better understand the translational gap, I believe that the article could be improved by adding summarized information regarding clinical trials of RIC for CCH and ischemic stroke conducted so far.

6.   [Figure] Figure 1 is not consistent with the main text. For instance, CBF improvement is listed as one of the neuroprotective mechanisms of RIC in acute ischemic stroke in Figure 1; however, it is not clearly described in the main text.

Author Response

Reviewer: 1

Comments to the Author

In this article, the authors reviewed the literatures regarding remote ischemic conditioning (RIC) for chronic cerebral hypoperfusion (CCH) and ischemic stroke, summarizing their biological mechanisms. I find this review article to be well-organized and easily comprehensible even for non-specialists in this field. However, I would like to point out a few things to enhance the value of this article.

Comment 1: [Overall] In this article, the authors described the mechanisms of RIC, the combinatorial effects of RIC and other treatments, and the consequent future directions, referring both pre- and post-conditioning. It may be applicable to discuss the mechanisms by combining insights from pre-conditioning and post-conditioning in chapters 2 and 3, as there are overlapping parts in their mechanisms. However, from a translational perspective, they should be clearly distinguished in chapter 6. For example, applying pre-conditioning to acute ischemic stroke is challenging; however, post-conditioning could be applied with the expectation of therapeutic effects.

Response 1 Based on your suggestions, we have made detailed revisions and additions to Chapter6, "Challenges and Future Directions," to more clearly distinguish the different applications and challenges of pre-conditioning and post-conditioning in clinical translation. we have made the following revisions to our manuscript.

Pre-conditioning and post-conditioning are two strategic applications of RIC that offer distinct opportunities for neuroprotection in the context of cerebrovascular disease. Pre-conditioning involves the application of RIC prior to the onset of ischemia, aiming to prepare the brain for a potential ischemic event by inducing a state of tolerance. This approach has been shown to be effective in experimental models, where the timing of the ischemic event is controlled. Post-conditioning, on the other hand, is applied after the onset of ischemia and takes advantage of the brain's innate plasticity during the critical period following a stroke. This strategy has the potential to modulate the ischemic response and promote restorative processes, offering a more clinically feasible window for intervention.

Comment 2: [Neuroprotective mechanisms of RIC on stroke] The authors described the reduction of excitotoxicity as a mechanism of RIC for acute phase ischemic stroke. However, the current article does not provide direct evidence suggesting that the reduction of excitotoxicity is the mechanism underlying acute phase neuroprotection in ischemic stroke. Reference #51 used different animal models; therefore, the authors should cite other appropriate references to support this mechanism.

Response 2Thank you for your insightful comment. We have checked carefully and supplemented the appropriate content and references in the manuscript.

Comment 3: [Neuroprotective mechanisms of RIC on stroke] Regarding chapter 3.2, Chronic Recovery Phase Neuroprotection, some references do not provide mechanistic evidence of RIC for ischemic stroke in the chronic phase. For instance, references #85 and #89 reported results related to angiogenesis or neurogenesis up to 14 days after MCAO and RIC, which do not support neuroprotective mechanisms in the chronic phase. Additionally, since reference #17 utilized animal models for CCH, citing this study to explain the mechanism of RIC for chronic ischemic stroke may be somewhat confusing.

Response 3we acknowledge your concern about the distinction between the subacute and chronic phases in the context of RIC's neuroprotective mechanisms post-MCAO. We have reviewed the literature and have clarified that while the subacute phase is typically considered to span from a few days to up to 14 days post-ischemia, the chronic phase extends beyond this period, often defined as starting around 7 days and lasting for several weeks to months thereafter. In our manuscript, both of these studies focus on the effects of RIC during the chronic recovery phase of stroke. They demonstrate that RIC can promote arteriogenesis and neurogenesis14 days post-stroke, providing a foundation for the improvement of long-term neurological function. These findings are validated within the articles, supporting the notion that RIC has protective effects during the chronic recovery phase of stroke. Additionally, we identified another study showing that RIC enhances angiogenesis and neurogenesis14 days after stroke, leading to improved neurological function recovery by day26.

Comment 4: [Overall] The abbreviation of ‘2VO’ for bilateral carotid artery occlusion may be a little confusing.

Response 4We sincerely thank the reviewer for careful reading. The 2-vessel occlusion (2VO) model is an animal model of chronic cerebral ischemia prepared using permanent ligation of the bilateral common carotid arteries, causing a state of chronic cerebral hypoperfusion. Since the2VO model involves the occlusion of both common carotid arteries, many studies use "2VO" as an abbreviation for this procedure to reflect the dual vessel involvement. We have corrected the abbreviation "2VO" in the manuscript to ensure clarity for all readers.

Comment 5: [Challenge and Future Directions] To better understand the translational gap, I believe that the article could be improved by adding summarized information regarding clinical trials of RIC for CCH and ischemic stroke conducted so far.

Response 5We greatly appreciate your suggestion to include summarized information regarding clinical trials of RIC for CCH and ischemic stroke to bridge the translational gap. We concur that this information would be beneficial for readers to understand the practical applications of RIC.

As you noted, we have indeed provided a brief overview of the clinical outcomes of RIC in the Introduction section, highlighting its potential and the need for further exploration. The focus of our review is primarily on the biological mechanisms underlying RIC, as we aim to offer a comprehensive synthesis of the current mechanistic understanding in preclinical studies.

However, to address your suggestion and to provide a more complete picture, we have added a concise summary of the clinical trials in the final section of our manuscript, under "Challenges and Future Directions." This addition briefly mentions the clinical studies, emphasizing the safety and potential benefits observed with RIC in patients with acute ischemic stroke and those with intracranial atherosclerosis.

Comment 6. [Figure] Figure 1 is not consistent with the main text. For instance, CBF improvement is listed as one of the neuroprotective mechanisms of RIC in acute ischemic stroke in Figure 1; however, it is not clearly described in the main text.

Response 6We feel sorry for our carelessness. In our resubmitted manuscript, we have revised Figure 1.

Reviewer 2 Report

Comments and Suggestions for Authors

The authors of this review attempt to thoroughly describe the neuroprotective mechanisms of remote ischemic conditioning, categorizing its effects during the acute and chronic phases of stroke recovery. I find this topic very interesting and gaining more and more publicity. The manuscript is prepared in accordance with the applicable rules for preparing scientific papers. The introduction provides a very broad scientific introduction to the topic. The manuscript thoroughly describes ischemic changes at the molecular level based on the available literature.

The added value is the final chapter describing possible further development prospects of this interesting research direction.

I have a small comment regarding the graphic design of the manuscript. I suggest slightly modifying the tables so that they become clearer and more legible for the potential reader.

I congratulate the authors on an interesting paper.

Author Response

Comment1: The authors of this review attempt to thoroughly describe the neuroprotective mechanisms of remote ischemic conditioning, categorizing its effects during the acute and chronic phases of stroke recovery. I find this topic very interesting and gaining more and more publicity. The manuscript is prepared in accordance with the applicable rules for preparing scientific papers. The introduction provides a very broad scientific introduction to the topic. The manuscript thoroughly describes ischemic changes at the molecular level based on the available literature.

The added value is the final chapter describing possible further development prospects of this interesting research direction.

I have a small comment regarding the graphic design of the manuscript. I suggest slightly modifying the tables so that they become clearer and more legible for the potential reader.

I congratulate the authors on an interesting paper.

Response1Thank you for your valuable feedback on our manuscript. We have revised the tables to make them more clearer.